# From Anti-PD-1/PD-L1 to CTLA-4 and to MUC1—Is the Better Response to Treatment in Smokers of Cancer Patients Drug Specific?

**DOI:** 10.3390/jpm11090914

**Published:** 2021-09-13

**Authors:** Lishi Wang, Fengxia Liu, Jing Li, Li Ma, Helin Feng, Qingyi Liu, William C. Cho, Haiyong Chen, Hong Chen, Hua Guo, Zhujun Li, Scott C. Howard, Minghui Li, Baoen Shan, Weikuan Gu, Jiafu Ji

**Affiliations:** 1Department of Basic Medicine, Inner Mongolia Medical University, Jinshan Development Zone, Huhhot 010110, China; Lwang37@uthsc.edu (L.W.); lizhijunmail@sina.com (Z.L.); 2Department of Orthopaedic Surgery and Biomedical Engineering, The University of Tennessee Health Science Center, Memphis, TN 38103, USA; jli115@uthsc.edu (J.L.); manymary@126.com (L.M.); fenghelin0311@126.com (H.F.); 3Research Center, The Fourth Hospital of Hebei Medical University, 12 Jiankang Road, Shijiazhuang 050011, China; lfx135246@sina.com (F.L.); zhongmeijian-lqy@163.com (Q.L.); baoenshan@hbydsy.com (B.S.); 4Beijing Cancer Hospital and Key Laboratory of Carcinogenesis and Translational Research, Department of Gastrointestinal Surgery, Peking University Cancer Hospital and Institute, Beijing 100142, China; 5Center of Integrative Research, The First Hospital of Qiqihar, Qiqihar 161005, China; 6Affiliated Qiqihar Hospital, Southern Medical University, Qiqihar 161007, China; 7Department of Clinical Oncology, Queen Elizabeth Hospital, Kowloon, Hong Kong, China; chocs@ha.org.hk; 8School of Chinese Medicine, The University of Hong Kong, Pok Fu Lam, Hong Kong, China; haiyong@hku.hk; 9Heilongjiang Academy of Traditional Chinese Medicine, 41 Xiangshun St, Xiangfang District, Harbin 150040, China; zyykxych@163.com; 10Department of Pathology and Cell Biology, College of Physicians and Surgeons, Columbia University Medical Center, New York, NY 10032, USA; hg2489@cumc.columbia.edu; 11Department of Acute and Tertiary Care, University of Tennessee Health Science Center, Memphis, TN 38103, USA; showard5@uthsc.edu; 12Health Outcomes and Policy Research, Department of Clinical Pharmacy and Translational Science, University of Tennessee College of Pharmacy, 881 Madison Avenue, Room 219, Memphis, TN 38163, USA; mli54@uthsc.edu; 13Research Service, Memphis VA Medical Center, 1030 Jefferson Avenue, Memphis, TN 38104, USA

**Keywords:** cancer, clinical predictor, drug, hazard ratio, lung, smoking

## Abstract

Whether smokers respond to anti-cancer drugs differently than non-smokers remains controversial. The objective of this study is to explore whether the better response of the smokers is specific to therapy of anti-PD-1/PD-L1, anti-checkpoint inhibitor, individual drugs on the cell surface, or lung cancer. Our results showed that among all non-small cell lung cancer (NSCLC) patients, when the data from anti-PD-1/PD-L1, anti-CTLA-4, and anti-MUC1 drugs are combined, the mean hazard ratios (HR) of smokers and non-smokers were 0.751 and 1.016, respectively. A meta-analysis with a fixed effect (FE) model indicated that the smokers have an HR value of 0.023 lower than that of the non-smokers. A stratified subgroup meta-analysis indicated that when treated with anti-CTLA-4 drugs, smokers had reduced HR values of 0.152 and 0.165 on average and FE model meta-analysis, respectively. When treated with an anti-MUC1 drug, smokers had reduced HR values of 1.563 and 0.645, on average and FE model meta-analysis, respectively. When treated with a combination of nivolumab and ipilimumab drugs, smokers had, on average, reduced HR and FE model meta-analysis values (0.257 and 0.141), respectively. Smoking is a clinical response predictor for anti-PD/PD-L1 monotherapy or first-line treatment in lung, urothelial carcinoma, and head and neck cancer. Smokers treated with other drugs have shown worse responses in comparison to non-smokers. These data suggest that, along with the progress in the development of new drugs for cancer, drugs acting on specific genotypes of smokers likely will arise.

## 1. Introduction

Traditionally, it has been recognized that smoking increases the risk for enhanced disease progression and affects the efficacy of treatment in patients with non-small cell lung cancer (NSCLC). Data from the Centers for Disease Control and Prevention (CDC) show that smoking is linked to about 80% to 90% of lung cancers (https://www.cdc.gov/cancer/lung/basic_info/risk_factors.htm, accessed on 22 April 2020). In a prior analysis of patients treated for NSCLC, we found that non-smokers had a lower risk of 0·16 HR compared to those who had ever smoked, with only a single study demonstrating conflicting results [1]. However, recent results from several large clinical trials of programmed death-1 (PD-1) and programmed death-ligand 1 (PD-L1) inhibitors showed that smokers had lower HR values than non-smokers [2,3,4,5]. Wallis et al. (2019), while conducting a meta-analysis on the association between smoking and survival of patients having benefited from immunotherapy in advanced malignancies, concluded that there was no difference between smokers and non-smokers in response to immunotherapy. In the same sense, Mo et al. (2020) reported that smokers benefited from either anti-PD-1/PD-L1 monotherapy or the combined regimen compared with chemotherapy. These contrasting data raise the question of the mechanism underlying the differential response of smokers treated with PD-1/PD-L1 and potentially other immunotherapy drugs.

PD-1 is an important factor in the normal immune response to prevent autoimmunity. Currently, clinical trials have reported results from five drugs in this category: two anti-PD-1 drugs (Nivolumab and Pembrolizumab) and three anti-PD-L1 drugs (Atezolizumab, Durvalumab, and Avelumab). Nivolumab, a fully humanized monoclonal antibody against PD-1, has shown a survival benefit in a number of cancers, including malignant melanoma [6], NSCLC [7], advanced renal cell carcinoma [8], and other types of cancers [9]. Pembrolizumab was the first anti-PD-1 antibody to be approved by the US Food and Drug Administration (FDA) for the treatment of patients with unresectable or metastatic melanoma with disease progression following Ipilimumab. It has also been used for the treatment of NSCLC [10] and other cancers. Atezolizumab is a PD-L1 inhibitor used in cancer therapy with a focus on bladder and NSCLC [11]. Durvalumab is reported as a selective, high-affinity, human IgG1 monoclonal antibody that blocks PD-L1. It has been used for the treatment of NSCLC [12] and urothelial carcinoma [13]. Avelumab is a promising new therapeutic agent for patients with metastatic Merkel cell carcinoma that has also been used in the treatment of NSCLC [14].

CTLA-4 is the second molecule of the immune checkpoint that has been targeted by monoclonal drugs [15]. Data on the status of smokers in response to the anti-CTLA-4 drug treatment are available. The anti-CTLA drug, Ipilimumab was efficient intreating NSCLC [16,17]. Both anti-PD-1/PD-L1 and CTLA-4 are also cell surface molecules [16,17,18,19,20,21]. The question is whether smokers respond better to both checkpoint drugs and their combination. 

Furthermore, the question remains on whether smokers respond better only to checkpoint drugs, immune stimulators, or modulators on the cell surface which include checkpoint drugs. MUC1, as an indicator for the presence of adverse conditions in the extracellular environment, has not been recognized as a molecule of immune checkpoints but instead as a cell surface regulator molecule [22]. Data on the status of smokers in two clinical trials using the anti-MUC1 drug, Tecemotide, are available [23,24]. These data can be used to compare other drugs such as anti-EGFR [25].

The objective of this study is to determine the efficacy of PD-1 and PD-L1, UTLA-4, MUC1, and EGFR drugs in the treatment of smokers and non-smokers among NSCLC and other cancer patients. The protocol has been registered at the International Prospective Register of Systematic Reviews (PROSPERO) with the registration number: CRD42019146402.

## 2. Methodology and Criteria

### 2.1. Search Strategy and Selection Criteria

A systematic review and network meta-analysis have been used to analyze the response of smokers to drug treatment in cancer patients in accordance with the Preferred Reporting Items for Systematic Reviews and Meta-Analyses [26]. Ethical approval was not required for this study since it involved data analysis. Some of the key points for the systematic and meta-analysis are provided below [27].

Data for the review were selected from clinical journals by searching major databases of publications such as PubMed, PMC/MEDLINE, EMBASE, and Scopus journals. The full texts and data were independently examined in duplicate by multiple authors for selection based on agreed criteria.

The title and abstracts of each article were screened for relevance by LW or WG. The full texts and data were independently examined with crosschecking by three authors (JL, LM, and FL) for selection based on agreed criteria. The final list of inclusion was determined based on discussion among authors. There was no disagreement on the list of collections.

Publication searching methods and initial inclusion and exclusion criteria were restricted to Clinical trials having a total patient number of more than 100 and with information on the smoking status of patients. Only articles published in English after 30 June 1990, and up to 1 December 2020, were included. Literature searches were conducted between 15 January and 1 July 2019. A final check on the updated articles was conducted on 24 December and 30 December 2020, for potentially missed articles. For the collected articles, we compared treatment efficacy among patients having a different smoking status in clinical trials for a variety of cancers.

### 2.2. Data Extraction

Data on safety risk to patients was used as the measure of response to drug treatment. Measurement of response to treatment was based on hazard ratios (HRs) for progression-free survival (PFS) and overall survival (OS) following drug treatment of different cancers [1]. The HR of the PFS/OS ratio has been used for the measurement of drug efficacy in clinical trials [1] while in this review they are used as indicators of a better (low HR) or a worse (high HR) response to drug treatment. The values or information on HR of the PFS/OS ratio were searched throughout the main text as well as the appendix of the selected articles. For each article, the main text was searched first. If the PFS or OS of patients and their smoking status were not found, the Appendix A/Appendix were then searched.

In general, either PFS or OS was provided from the studies. A few reports provided both PFS and OS or multiple PFS from analyses of multiple sub-groups. In these cases, both PFS and OS or multiple PFS were analyzed with proportionally reduced population size or divided by the number of subgroups, based on the information of subgroup analyses. Whenever the data for more than one HR pair were included in a single study among the collected data, both continued fixed-effects (FE) and random-effects (RE) models were analyzed for all the data. *p*-values ≤ 0.05 were regarded as significant between samples from smokers and nonsmokers. RE models were used under the assumption that underlying true effects differ across trials given the heterogeneity of populations from this variety of clinical trials. For each step, analyses were conducted by two persons independently to confirm the results.

### 2.3. Quality Assessment

Each article was reviewed by three researchers independently for the criterion based on CONSORT compliance, and journal credentials (Appendix A). This included the randomizing of participants, of patient basic demographic and diagnostic characterization, of data collection procedures, and of collectors. In addition, the phase of clinical trials, the sample sizes of smokers and non-smokers, potential age and sex influence on response to treatment by smokers and non-smoker, and journal credentials are also used as the selection criteria. Reviewers resolved all disagreements by discussion without the arbitrator.

### 2.4. Data Validation with Corresponding Authors of the Studies of PD-1/PD-L1 Drugs

For HR data on smokers and non-smokers collected from each of the articles, a data confirmation email was sent to the corresponding authors of the publications. The content of the email included the following requests: (i) provide information on what data we collected from the authors’ publication, (ii) ask the author to confirm the collected data; (iii) ask if any additional or detailed information is available; (iv) ask whether any significant factor might have influenced HR values of smokers vs. non-smokers in the study. For large clinical trials that did not provide HR values of smokers vs. non-smokers, an information request email was sent to the corresponding authors (Appendix A). The email provided: (i) the title of the author’s article; (ii) the purpose of data collection by our group, and requested; (iii) whether the author has HR data on patients with different smoking statuses, and (iv) willingness for collaboration and sharing and use of the data requested. In order to make the email different from commercial emails, the title was “Information regarding your publication: ‘Journal name, Year, Page numbers’” for the data collection emails and “Data in your publication: ‘Journal name, Year, Page number’ for the data confirmation/collaboration request emails. 

### 2.5. Meta-Analysis

#### 2.5.1. Meta-Analysis

A meta-analysis was conducted with a procedure from our previous publication [1] and followed the PRISMA statement for reporting systematic reviews. Data preparation followed our previous method [1]. The analysis was conducted among different categories between smokers and non-smokers, including drugs for NSCLC and other cancers. 

#### 2.5.2. Stratified Meta-Analysis

In each category, a stratified meta-analysis was conducted between smokers (current and former together) vs. non-smokers, current smokers vs. non-smokers, and former smokers vs. non-smokers, wherever the data qualified for such analyses. For the comparison between smokers and non-smokers, the HR values of current and former smokers were combined into one balanced HR value using the following equation of normalization: HR (value of smokers) = {HRc*Nc/(Nf + Nc) + HRf*Nf/(Nf + Nc)}, where HRc = HR values of current smokers, HRf = HR value of former smokers, Nc = number of current smokers, and Nf = number of former smokers [10].

#### 2.5.3. Bias Analysis

We followed our procedure for the bias analysis from our protocol [26]. In general, either PFS or OS was provided from the studies. A few reports provided both PFS and OS or multiple PFS from analyses of multiple sub-groups. In these cases, both PFS and OS or multiple PFS were analyzed with proportionally reduced population size or divided by the number of subgroups, based on the information of subgroup analyses. Whenever the data for more than one HR pair were included in a single study among the collected data, both continued fixed-effects (FE) and random-effects (RE) models were analyzed for all the data. *p*-values ≤ 0.05 were regarded as significant between samples from smokers and nonsmokers. RE models were used to define a priori given the heterogeneity of populations from this variety of clinical trials. For each step, analyses were conducted by two persons independently to confirm the results.

## 3. Smoking as a Favorite Predictor for Drugs of Anti PD-1/PD-L1, MUC1, CTLA-4

The effects of smoking on treatment outcome data were analyzed with a meta-analysis for two major categories of drugs, immune checkpoint inhibitors (anti-PD-1/PD-L1 and CTLA-4 drugs) and other drugs including MUC1, EGFR, and angiogenesis inhibitors. Diseases treated with anti-PD-1 and anti-PD-L1 drugs included NSCLC and other types of cancers. Data analyzed for other drugs were only from NSCLC patients. To examine the effect of only anti-PD-1 and anti-PD-L1, within the data of anti-PD-1 and anti-PD-L1 drugs, a sub-group analysis was conducted based on the treatment matrix, including monotherapy versus combination treatment and maintenance therapy. To examine the effect of combination between anti-PD-1/PD-L1 and CTLA-4 drugs, patients’ responses to the combination treatment between drugs from these two types of checkpoint drugs were analyzed. 

Our review is based on published results from clinical trials from PubMed, PMC/MEDLINE, and Scopus. Inclusion criteria were reported in clinical trials with different smoking statuses. Table 1 shows data from a set of 36 clinical trials including 13 on NSCLC and 6 on other cancers treated with PD-1/PD-L1 drugs, 5 on cancer patients treated with anti-CTLA-4 drugs, 2 treated with anti-MUC1 drugs, and 10 with anti-VEGF drugs.

For anti-PD-1/PD-L1 drugs, the mean HR value from the smoking subgroups in these trials was 0.702; non-smoking subgroups was 0.848. When we combined the anti-PD-1/PD-L1 and anti-CTLA-4 drugs, the mean HR values of smokers and non-smokers were 0.735 and 0.882, respectively. When we combined the anti-PD-1/PD-L1, anti-CTLA-4 drugs, and anti-MUC1 drugs, the mean HR values of smokers and non-smokers were 0.751 and 1.016, respectively. The *p*-values for comparison between smokers and non-smokers for data of three types of drugs, two types of drugs, and anti-PD-1/PD-L1 drugs only are 0.0325, 0.0299, and 0.0685, respectively (Figure 1A). The meta-analysis indicated that the smokers have an HR value of 0.023 lower than that of the non-smokers (Figure 1B) with heterogeneity of I2 = 0% and *p* < 0.001. In contrast, for anti-VEGF drugs, the mean HR value from the smoking subgroups in these trials was 0.868; the mean HR value from the non-smoking subgroup subgroups was 0.654, with a *p*-value of 0.0013 (Figure 1C). The meta-analysis indicated that the smokers have an HR value of 0.194 higher that of the non-smokers (Figure 1D), again with heterogeneity of I2 = 0% and *p* < 0.001. These data suggest a significant difference in the response to treatment between smokers and no-smokers when different drugs were used. Accordingly, a detailed description and subgroup and stratified meta-analysis were conducted to further analyze the difference between smokers and non-smokers for these drugs. 

### 3.1. Data from NSCLC Outcomes Following Treatment with Anti-PD-1 and Anti PD-L1 Drugs Suggest That Smokers Respond to Treatment Better than Non-Smokers

As indicated in the supplementary data, many studies with smaller sizes either do not have the data on smoking or are duplicates in the publications with a large number of patients. Data from a total of 13 clinical trials of NSCLC were collected for detailed analysis (Appendix A).

We conclude that when comparing current smokers with non-smokers, treatment with existing immune checkpoint inhibitors other than Durvalumab achieved a better response in smokers. However, these data also suggest that a better response is only observed when one of these treatments is administered singly without any other drug adjuvant therapy.

#### 3.1.1. Pembrolizumab

Five publications were included that reported results of clinical trials in lung cancer patients [4,27,28,29,30]. One had 38 patients, which is far less than the 100 patient minimum for our inclusion [27]. The smoking status of patients was not reported in one study [28]. 

Results from the remaining three clinical trials indicated that in PD-L1-Positive NSCLC patients, the effect of the drug in current smokers was better than that of the non-smoker (Appendix A). An early report from Garon et al. (2015) showed that the response to treatment in current and former smokers is greater than that of never smokers in a sample of 305 PD-L1-positive NSCLC patients [31]. In all three types of subgroup analyses, the HR values of current and former smokers were lower than the non-smokers ( Appendix A). Moreover, when compared following treatment with pembrolizumab and chemotherapy together, smokers had a lower mean HR than non-smokers. 

#### 3.1.2. Nivolumab

Five studies were phase 3 with smoking status analyzed [18,32,33,34,35]. Among these five studies, [32,33] used fewer patients. These four studies, [2,32,33,35], reported preliminary data from the studies of Borghaei et al. (2015), which contained a sufficient number of never-smoker patients [34]. The study compared the effect of Nivolumab to a non-PD-1 drug, Docetaxel, in patients treated for advanced squamous- NSCLC [36]. The mean HR value of never smokers was higher than that of both current and former smokers in response to Nivolumab (Appendix A). 

An exception is a study on 67 patients with nine nonsmokers. The results indicated that nonsmokers fared worse than former and current smokers in both PFS (1.93 vs. 0.85 and 1) and OS (2.15 vs. 1.09 and 1) [37].

From Scopus, an article by Carbone et al. on the study of stage IV of NSCLC was added [38].

#### 3.1.3. Atezolizumab

Data from these four studies [39,40] were used for the analysis. It is important to note that Socinski et al. (2018) reported that Atezolizumab improved PFS and OS among patients with metastatic nonsquamous NSCLC, regardless of PD-L1 expression and EGFR or anaplastic lymphoma kinase (ALK) genetic alteration status. In their study (Socinski et al. (2018)), the mean PFS of never smokers was higher than that of smokers (Appendix A). 

More recently, West et al. (2019) reported the results of a trial of Atezolizumab in combination with carboplatin plus nab-paclitaxel chemotherapy [41]. The data indicated that the smokers had a higher mean HR value than the non-smokers. 

#### 3.1.4. Avelumab

One recent publication on a phase 3 clinical trial of NSCLC patients examined smoking status. Similar to the results described above, in the PD-L1-positive population, the mean HR of OS in smokers was much lower than that of non-smokers [2]. 

#### 3.1.5. Durvalumab

Durvalumab differs from the four drugs reported above in that it is a human immunoglobulin G1 kappa monoclonal antibody that blocks the interaction of PD-L1 with PD-1. It also blocks the interaction of PD-L1 with CD80 molecules.

Three recent publications on clinical trials phase 3 in NSCLC patients were obtained. One by Antonia et al. (2017) contained data on smoking status. When patients previously exposed to anti-PD-1 or PD-L1 antibodies (assuming these are PD-1-positive individuals) were excluded, PD-L1 expression of 25% or more on tumor cells occurred in 22.3% of patients, and the mean HR value of PFS in non-smokers was much lower than that of smokers [42]. Further searching with the keywords “Durvalumab phase-III” yielded 36 publications. One additional follow-up publication by the same group [43], the PACIFIC Investigators, reported OS data from the same study. Like their previous report, the mean OS in non-smokers was better than that of smokers (Appendix A). However, that treatment followed chemoradiotherapy, not treatment with Durvalumab alone.

#### 3.1.6. Meta-Analysis of Significant Differences in Distribution Pattern of HR Values between NSCLC Patients Treated with the PD-1/PD-L1 Inhibitors

We first compared HR values of smokers and non-smokers of NSCLC patients treated with PD-1/PD-L1 (Appendix A). For the anti-PD-1/PD-L1 drugs, data from a total of 13 studies are included. The meta-analysis compared responses to treatment between NSCLC smokers and non-smoker patients (Figure 2). The forest plot clearly showed a significantly different distribution of data from different studies. Thus, while some studies showed a better response to treatment for smokers, others showed the opposite result. The meta-analysis using a fixed-effects (FE) model produced a *p*-value of <0.001. The smoker group was estimated to have an HR of 0.037 lower than that of the non-smoker group (Figure 2A), with heterogeneity I2 = 0% [1]. The leave-one-out model produced the same HR reduction in smokers of 0.037 in comparison with that of non-smokers (Figure 2B). Further analyses using a random-effects (RE) model also produced a *p*-value of <0.001, supporting the significance of the difference (Figure 2C). The random-effects model suggested that the smoker group had a 0.098 lower HR value than that of non-smokers. However, the I2 value was 97.35%, suggesting potential heterozygosity among samples. Therefore, further analysis with the leave-one-out model was conducted. The leave-one-out model of RE produced the same HR value (Figure 2D). These data indicate that although heterozygosity may exist among these samples, the difference between smokers and non-smokers is most likely true. Thus, overall, there was a significant difference between smokers and non-smokers in HR values for patients having drug treatment of lung cancer with PD-1/PD-L1 drugs. 

### 3.2. Stratified Meta-Analysis of PD-1/PD-L1 Drugs on NSCLC

Because of the significant differences among data from the treatments with PD-1/PD-L1 drugs between smokers and non-smokers, we conducted a stratified meta-analysis (Table 1; Appendix A). The first grouping is between the first-line treatment and the second-and third-line treatment with or without chemotherapy [4,29,39,41,42]. A stratified meta-analysis of PD-1/PD-L1 drugs on NSCLC between smokers and non-smokers is shown in Figure 3. Our analysis indicated that smokers in the first-line treatment group responded better than non-smokers [4,29,34,39] with a 0.309 HR value reduction and with heterogeneity I2 = 0% and *p* < 0·001 (Figure 3A). While in the second-and third-line treatment, or in combination with other drugs, non-smokers fared better than smokers (Figure 3B) [36,39,40].

Our second comparison was between monotherapy and monotherapy combined with chemotherapy and the case of multiple immunotherapies. In this case, the smokers treated with monotherapy or mono-plus chemotherapy fared better than those treated with multiple therapies (Figure 3C,D) [2,34,40]. In the case of monotherapy, smokers had an HR value of 0.164 lower than that of non-smokers. The analysis showed heterogeneity I2 = 0% and *p* < 0.001 while the combined therapy indicated that non-smokers had a lower mean HR value than smokers (Figure 3D) [41,42]. 

Together, there were six studies of first-line treatments and monotherapy identified [2,4,29,34,39,40]. Among them, the study of Mok et al. (2019) provided three HR values, which were separately analyzed in the stratified meta-analysis. The total population, therefore, was divided by three because of these three HR values to adjust for their weight in the analysis. The mean HR value of smokers was 0.218 lower than that of the non-smokers (Figure 3E). While the combination of second/third-line treatment and multiple drug treatments from a total of five studies indicated that the mean HR value of smokers was 0.156 higher than that of non-smokers (Figure 3F). Both of these analyses showed heterogeneity I2 = 0% and *p* < 0.001 (Figure 3D,F) [36,41,42,43]. The stratified meta-analysis with leave-one-out and RE models confirmed the results (Appendix A).

### 3.3. Response to Treatment by Smokers and Nonsmokers Treated with Anti-PD-1 PD-L1 Drugs in Other Cancers

Responses to treatment by PD-1/PD-L1 drugs were searched for clinical trials for other six cancers, including renal [2], head-and-neck, gastric or gastro-esophageal, hepatocellular carcinoma, melanoma [3], and urothelial (Appendix A). Two of these studies reported the smoking status of the patients by pembrolizumab [44,45]; smoking status was reported in the study of head and neck cancers by nivolumab. The HR for OS in non-smokers was lower than that of smokers [46,47]. One publication on the treatment of urothelial carcinoma by atezolizumab was identified, which reported smoking status [48]. Again, current smokers had a lower mean HR than non-smokers. Two publications for renal-cell carcinoma were identified but only one reported smoking status. Again, the mean HR of smokers was lower than that of non-smokers [49].

For each disease reporting treatment by PD-1/PD-L1 in Figure 4, we searched data of smokers and non-smokers treated by other drugs. For more than two eligible publications, we limited the maximum number for each disease to two trials, using whichever of these reported HR values of smokers and non-smokers.

Similarly, among patients with other cancers treated with drugs other than PD-1/PD-L1, the non-smokers responded better than the smokers (Appendix A).

Data from a total of six studies were collected for the meta-analysis. Three of them were monotherapy or first-line treatment [45,47,48], while three were second-line or used a combination of drugs [44,46,49] (Appendix A). The meta-analysis indicated that the mean HR value for smokers ranged between 0.028 and 0.038 lower than that of the non-smokers in the FE and RE model (Figure 4). Although the random effects model indicated that heterozygosity may influence the data (Figure 4C), the leave-one-out model showed the same results (Figure 4D). However, the sample size was relatively small. 

### 3.4. Additional Reports with Small Numbers of Patients Showing Smokers Responded Better than Nonsmoker in Patients Treated with Anti-Pd-1 and Anti-Pd-L1 Drugs

In view of the interesting results from the above publications, we extended our search for any additional “clinical” trials that provided the smoking status of patients. 

#### 3.4.1. Pembrolizumab

We found only one report that included smoking status, which showed that former smokers fared better than non-smokers [50]; In addition, we found an updated report from the KEYNOTE-024 study by [51] that showed similar results for HR for OS, i.e., that current and former smokers responded better to treatment than non-smokers (Table 1).

#### 3.4.2. Nivolumab

The objective response rates of smokers and non-smokers reported by [52] were added. As previously predicted, because the study subjects were NSCLC patients who failed prior platinum-based chemotherapy, and not first-line or PD-L1-positive, the response by smokers did not differ from that of non-smokers. The other data were from [37]. As mentioned above in the reports for anti-PD-1 and anti-PD-L1, the data were not included earlier due to the small number of patients. The data from this study were from NSCLC patients treated with nivolumab. The response to the treatment of smokers was better than the non-smokers [37] (Table 1).

#### 3.4.3. Atezolizumab and Avelumab 

From the database, no new data were found from these two drugs.

#### 3.4.4. Durvalumab

The response to treatment by Durvalumab for advanced NSCLC from non-smokers was reported as better than that of smokers in a study by ATLANTIC investigators [53]. However, Durvalumab was used as a third-line or later treatment. The complete analysis based on smoking status was then carried out by Sridhar et al. (2019). Their analysis again supports the notion that when Durvalumab was used as a first-line treatment, smokers responded better than non-smokers (Table 1).

### 3.5. Reports with Small Numbers of Patients Showing Smokers Responded Better than Nonsmoker in Patients Treated with Anti-CTLA-4 (Ipilimumab) Drugs

Data from five studies included the status of smokers and non-smokers [16,17,18,19,20]. Additional searches from PMC or Scopus did not add more data. By analysis of the data from these five studies, we found that when treated with anti-CTLA-4 drugs, smokers had a reduced HR value of 0.152 (HR 0.885, range from 0.57 to 1) in comparison to that of non-smokers (HR 1.037, range from 0.58 to 1.19) (Appendix A).

The meta-analysis indicated that overall smokers responded worse than non-smokers with a 0.011 HR value higher than the non-smokers and with heterogeneity I2 = 0% and *p* < 0.001 when the patients of NSCLC and non-NSCLC are included (Figure 5A,B) [16,17,18,19,20].

However, subgroup analysis on the response from NSCLC patients only indicated that smokers responded better than non-smokers with a 0.165 HR value less than that of the non-smokers, with heterogeneity I2 = 0% and *p* < 0.001 (Figure 5C,D).

Furthermore, by examining the data subgroup with a combination between anti-PD-1 (Nivolumab) and CTLA-4 (Ipilimumab) drugs, we found that smokers responded better than non-smokers with a 0.236 HR value less than that of the non-smokers, with heterogeneity I2 = 0% and *p* < 0.001 (Figure 5E,F) [18,19,20].

In contrast, for the subgroup with non-NSCLC patients and combination with other drugs, from the study of small cell lung cancer [17], and relapsed malignant pleural mesothelioma [20], smokers responded worse than that of the non-smokers by an increase of 0.068 in the HR value (Figure 5G,H). However, the combination between Nivolumab and Ipilimumab within this subgroup still had a lower HR value in smokers [20].

### 3.6. Reports with Small Numbers of Patients Showing Smokers Responded Better than Nonsmoker in Patients Treated with Anti-Mucin 1 (Tecemotide) Drugs

Data from two studies including the status of smokers and non-smokers [23,24] indicate that smokers exhibited much lower HR values, with a mean of 1.563 reduced HR lower than that of non-smokers. PMC/MEDLINE and Scopus did not reveal any additional data.

Figure 6 showed that in both FE and RE models, smokers had much lower HR values than that of non-smokers, with a 0.645 and 0.593 reduction, respectively. Because of the potential heterozygosity (Figure 6A) and small sample size, the data cannot be regarded as definitive. However, the chance of the same error in two studies with such a large number of samples is rare.

## 4. Smokers Response to the Treatment by Antibody Developed against EGFR and VEGF

Early analysis indicated that smokers respond worse than non-smokers to the treatments of several other types of drugs including anti-EGFR and VEGF in cancer patients (Appendix A) (1). The data on anti-PD-1/PD-L1, CTLA-4, and MUC1 drugs prompt a recheck response of smokers in cancer patients to these drugs. 

Our review supports the early conclusion on the drugs of anti-EGFR and VEGF. Response to Bevacizumab, an anti-VEGF drug is a typical example. Data from the 10 recent studies included the status of smokers and non-smokers [54,55,56,57,58,59,60,61,62,63]. Nine of them with information of the numbers of patients were used for the analysis [54,55,56,57,58,59,60,61,62]. Our analysis showed that in both FE and RE models, in contrast with the response from anti-PD-1/PD-L1, CTLA-4, and MUC1 drugs, smokers had a much higher HR value than that of non-smokers in response to treatment by anti-VEGF drugs, with 0.194 (Figure 1 and Appendix A). 

## 5. Conclusions Remarks and New Questions

### 5.1. Smoking Status and Treatment with Drugs Targeting on Molecules of Cell Surface in Cancer Patients

By examining data from these trials, we conclude that (1) during treatment of cancer patients with PD-L1 mutations/positive, current smokers are more likely to respond better than the non-smokers when treated with one of four of the five anti-PD-1 and PD-L1 drugs: Nivolumab, Pembrolizumab, Atezolizumab, and Avelumab. Such a response difference is not limited to NSCLC but is most likely to two other types of cancers, urothelial carcinoma, and head and neck cancer. (2) During treatment with the other two drugs, anti-CTLA-4 and anti-MUC1, smokers are more likely to respond better than the non-smokers in the treatment of NSCLC patients (Appendix A). The data for the drug of anti-CTLA-4 was from three out of five large clinical trials. The data for anti-MUC1 was from two clinical trials. We expect future clinical studies will confirm such a conclusion. (3) During treatment by multiple drugs or a combination of drugs, if drugs of anti-PD-/PD-L1 in combination with chemotherapy or the drug of anti-CTLA-4, smokers are more likely to respond better than the non-smokers. When combined with most other drugs, smokers are more likely to respond worse than non-smokers. However, it is expected that the combination among three drugs of anti-PD-/PD-L1, CTLA-4, and MUC1, smokers will respond better than non-smokers. (4) Data from drugs targeting VEGF showed that smokers respond worse than non-smokers, indicating a difference in response to treatments between smokers and non-smokers who are drug dependent.

Current data also clarified the combination of PD-1/PD-L1 inhibitors with other drugs. Thus, when PD-1/PD-L1 inhibitors are combined with CTLA-4 drugs, smokers are still responding to treatment better than nonsmokers. Only PD-1/PD-L1 inhibitors in combination with other drugs, e.g., VEGF, is the smoker response worse than that of non-smokers. In the second-line treatment with PD-1/PD-L1 inhibitors, smokers did not respond well, most likely due to the acquired resistance in the first treatment [64]. 

### 5.2. Clinical Implication

Based on our findings, we suggest: (1) it is important to consider a patient’s smoking status before embarking on anti-PD-1/PD-L1/UTLA-4/MUC1 treatment of NSCLC; (2) smoking status may be considered as a clinical predictor for the treatment of urothelial carcinoma, and head and neck cancer by PD-1/PD-L1 inhibitors. In future clinical trials or practice, smoking status should be taken into account for the other two types of drugs, the anti-UTLA-4 and anti-MUC1; (3) smoking status may be considered as a clinical predictor in the treatment with a combination of these three drugs; and 4) clinical smoking history is important in lung cancer, head neck, and bladder cancer because they are associated with squamous carcinoma. 

Smoking consideration in the cancer treatment agrees with several reported findings in pathogenic features: (1) Tumor subtype: squamous cell carcinoma. In NSCLC patients, smoking is well-known to be related to squamous cell carcinoma type [65]; (2) PD-L1 expression in tissue specimens. Squamous cell carcinoma type has a higher rate of PD-L1 positivity by IHC stain. Anti-PD/PD-L1 treatment response rate is highly associated with PD-L1 expression [66]; (3) tumor-infiltrating lymphocytes (TILs) is an important histology parameter when considering the predictor of anti-PD/PD-L1 treatment response. A study showed that CD8-positive TILs were more numerous in patients who had smoking-related KRAS mutation [67] and (4) tumor mutations: K-RAS mutation. Another study [68] showed that smoking is associated with higher tumor mutation. The KRAS mutation is associated more strongly with a history of heavy smoking and higher tumor mutation burden”. “Non-KRAS oncogene-driven lung cancer subtypes (e.g., ROS1, MET, BRAF, HER2, RET) is less well-studied, however, most of these other non-KRAS molecular subtypes (possibly apart from BRAF V600 mutation and MET gene alteration) are associated with a light-to-never smoking history [3,4,5]. In our study, most of the patients selected in the first-line treatment group were without a sensitizing EGFR mutation (except for a study by Socinski et al., 2018), while some patients in the second-and third-line treatment group were EGFR mutation-positive. Evidently, patients with EGFR mutations do not seem to benefit from immunotherapy (Appendix A).

### 5.3. New Research Questions

The most important question is whether smoking status is related to drug targeting at checkpoints or what types of targets. While PD-1/PD-L1 and CTLA-4 are key checkpoint targets, MUC1 has not been recognized to relate to the checkpoint. A common feature of these three targets is that they are cell surface markers or molecules. EGFR or VEGF are also cell surface proteins. Therefore, our hypothesis is that smoking changes some features of the cell surface; that the structures of some types of molecules have been altered, that some are not, therefore, the response to treatment between smokers and non-smokers is drug-specific or target-specific.

The next question is whether smoking alters features of the cell surface in the lung or in the whole body. Based on the fact that smokers respond better than non-smoker in the treatment by not only NSCLC but also urothelial carcinoma, and head and neck cancer by PD-1/PD-L1 inhibitors, we assume that smoking alters features of the cell surface in the whole body. Thus, if the smokers respond better in NSCLC by treatment of one drug, they will respond better in other types of cancer too by the same drug. 

Finally, what and how smoking alters the surface of the cell is a critical question for research. These questions will likely need to have a large amount of research before the mechanism of interaction between checkpoint inhibitors/cell surface markers and smoking will be understood. Potential new targets and therapeutic applications may also be discovered by such research.

### 5.4. Limitations

Based on the findings summarized here, this new paradigm needs a clear definition with additional studies. There are several conditions that define the better response from smokers. Mainly, the better response from smokers seems limited to the population having PD-1 mutation/positive expression for anti-PD-1/PD-L1 drugs. In studies with a large proportion of patients that were not PD-1-positive [27], non-smokers responded better than smokers [59,69]. Secondly, the better response of smokers occurred when patients were treated with anti-PD-1 and PD-L1 drugs alone or in combination with chemotherapy or anti-CTLA-4 drugs, not in combination with other drugs [39]—although the data from combination with MUC1 were not available. Thirdly, since most of the trials did not provide data separately for current and former smokers, it is not clear whether current smokers differ from former smokers, or, if former smokers may not respond as well as non-smokers. Two trials suggested that current smokers fared much better than former smokers [69], while one suggested the contrary [26,31].

We may have missed some data by not screening all the public resources, although the database and journals in this study are the most powerful databases and the most reliable medical journals. However, as we searched major clinical journals, we assumed that data from most large clinical trials were included. In addition, this review is based on data from a limited number of studies that provided the HR status of smokers and non-smokers. Due to the importance of such an issue, future studies may need to analyze the response to treatment of patients in a detailed category, such as current smokers, long-time smokers, and new smokers, longer-time former smokers, recently stopped smokers, non-smokers, and potential secondary smokers among the non-smokers.

We realize that the population sizes of non-smokers are rather small in most of the studies. However, the small sample sizes are not only for trials of anti-PD-1 and PD-L1 or MUC1 drugs. The small samples exist for other drug trials as well, while in these trials for MUC1, non-smokers responded better than smokers [59,69].

### 5.5. Final Remarks

It is critical to note that current evidence still stands on the concept that smoking leads to more damage to humans than being beneficial. More research is needed to clarify which and what types of drugs work on smokers better than the non-smokers. 

## Figures and Tables

**Figure 1 jpm-11-00914-f001:**
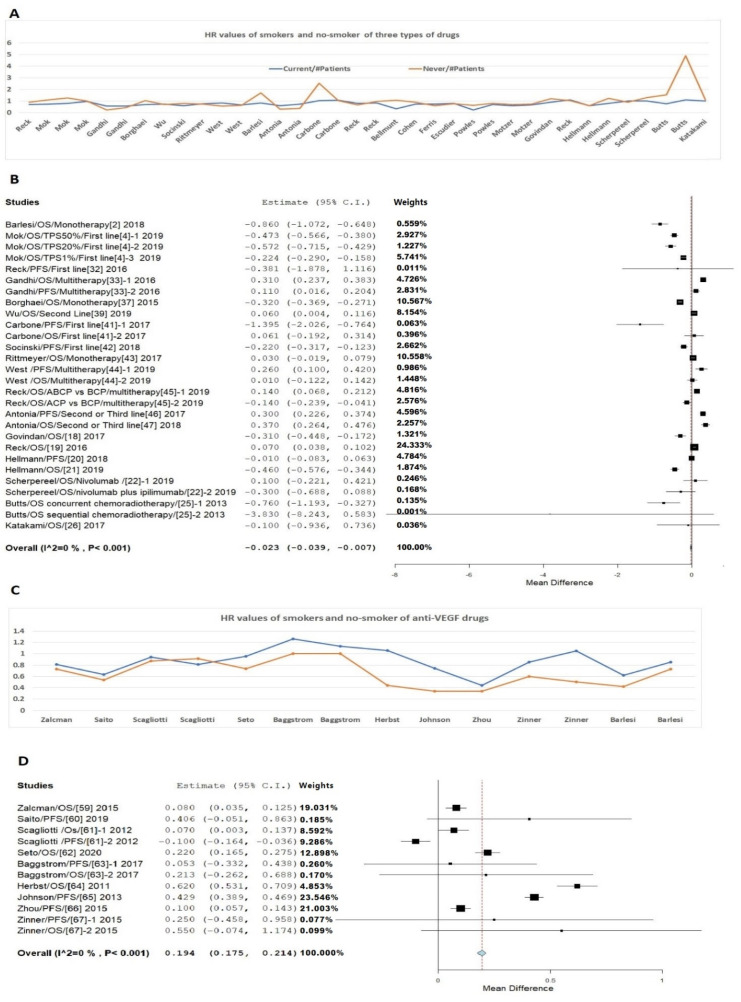
HR values of patients treated with different drugs. (**A**) HR data from three types of drugs (anti-PD-1/PD-L1, anti-CTLA-4 drug, and anti-MUC1 drug). (**B**) Meta-analysis of HR for these three types of drugs. (**C**) HR data from Bevacizumab on NSCLC between smokers and non-smokers. (**D**) Meta-analysis of HR Bevacizumab on NSCLC between smokers and non-smokers.

**Figure 2 jpm-11-00914-f002:**
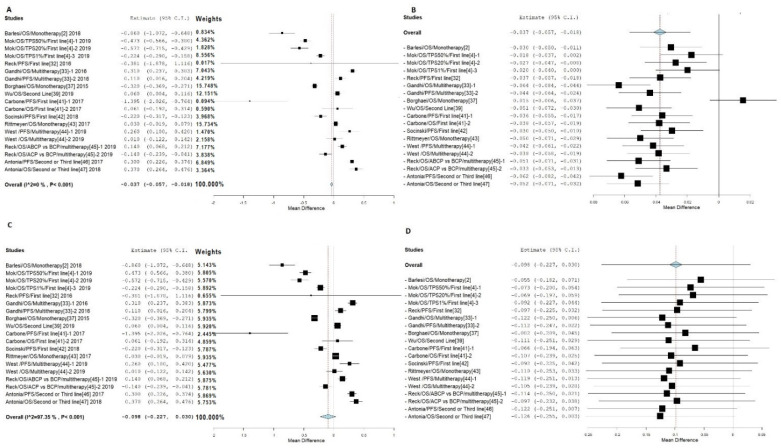
Meta-analysis of response to treatment compared between smoker and non-smoker NSCLC patients. (**A**) Shows results of the FE model with all sets of data. (**B**) Shows results of FE with the leave-one-out model. (**C**) Shows results of the RE model with all data. (**D**) Shows results of the RE model with the leave-one-out model.

**Figure 3 jpm-11-00914-f003:**
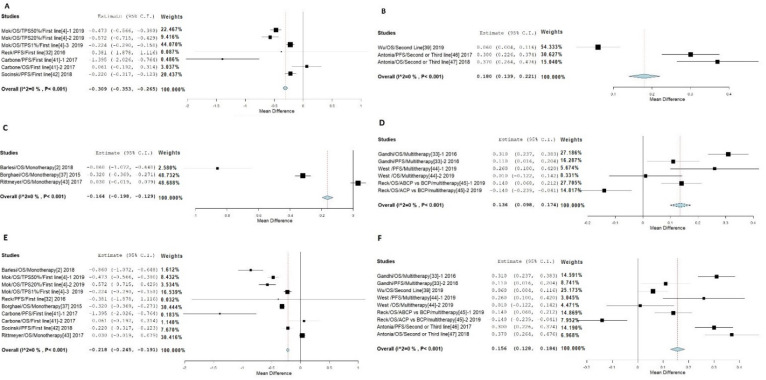
Stratified meta-analysis of PD-1/PD-L1 drugs on NSCLC between smokers and non-smokers. (**A**) First-line treatment. (**B**) Second- and third-line treatment. (**C**) Monotherapy. (**D**) Combined with other treatments. (**E**) Patients’ response when treated as first and monotherapy (RE). (**F**) Patients’ response when treated as second-line and multi-therapy (RE).

**Figure 4 jpm-11-00914-f004:**
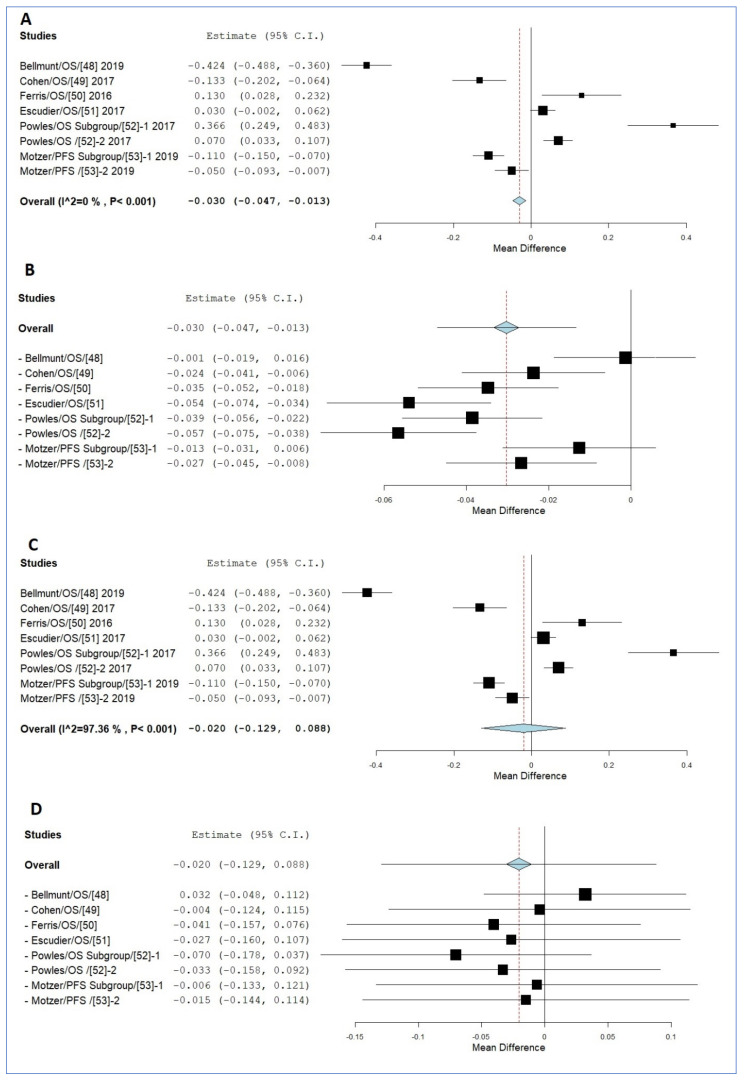
The results indicated that smokers responded better than non-smokers in other cancers. (**A**) Response to treatment by patients of different cancers (FE). (**B**) Response to treatment by patients of different cancers by the FE leave-one-out model. (**C**) Response to treatment by patients of different cancers by the RE model. (**D**) Response to treatment by patients of different cancers by the RE leave-one-out model.

**Figure 5 jpm-11-00914-f005:**
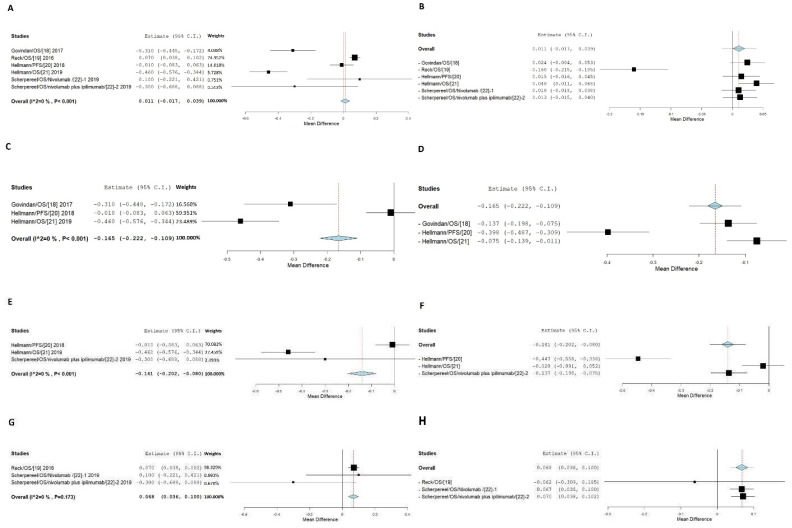
Meta-analysis of anti CTLA4 drugs on NSCLC and non-NSCLC between smokers and non-smokers. (**A**) Shows results of FE with all sets of data. (**B**) Shows results of FE with leave-one-out. (**C**) Shows results of FE with a subgroup of NSCLC patients. (**D**) Shows results of the FE model with leave-one-out for NSCLC patients. (**E**) Shows results of FE with a subgroup of combination between nivolumab and ipilimumab. (**F**) Shows results of the FE model with leave-one-out for combination between nivolumab and ipilimumab. (**G**) Shows the results of FE with non-NSCLC cancer or combination with other drugs. (**H**) Show the results of FE with leave-one-out for non-NSCLC or combination with other drugs.

**Figure 6 jpm-11-00914-f006:**
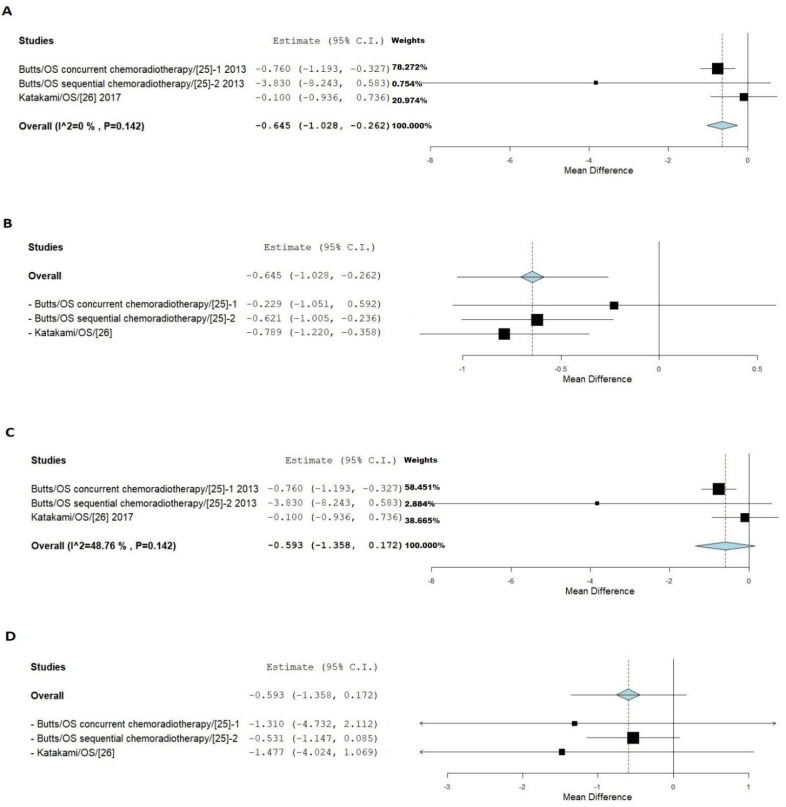
Meta-analysis of anti MUC1 drugs on NSCLC between smokers and non-smokers. (**A**) Shows results of FE with all sets of data. (**B**) Shows results of FE with leave-one-out model. (**C**) Shows results of RE with all sets of data. (**D**) Shows results of RE with leave-one out-model.

**Table 1 jpm-11-00914-t001:** Additional data collected from clinical trials.

Drugs/First Author	Drug Comparison	Subgroup Analysis	Current/# Patients	Former/# Patients	Never/# Patients	Overall/# Patients	Note
Pembrolizumab/Shaverdian (50)	KEYNOTE-001 phase 1 trial Former smokers vs non-smokers	Progression-free survival (PFS)	0.60 (0.38–0.95) vs. 1 *				
		Any previous radiotherapy and PFS	0.78 (0.47–1.31) vs. 1 *				
		Previous extracranial radiotherapy and PFS	0.82 (0.49–1.37) vs. 1 *				
Pembrolizumab/Reck (51)	Update on KEYNOTE-024 vs. Chemotherapy	Subgroup analysis of overall survival (OS)	0.81 (0.41–1.60)/65	0.59 (0.41–0.85)/216	0.90 (0.11–7.59)/24	0.63 (0.47–0.86)/305	All are PD-L1 expression on at least 50% of tumor cells
Nivolumab/Lee (52)	To patients failed prior platinum-based chemotherapy	Objective response rates	15/78 (19.2)		5/22 (22.7)		PD-L1 status not known.
Nivoluma/Dumenil (37)	Prospectively and treated by nivolumab in two French academic hospitals	PFS	1 (1.00–1.00) *	0.85(0.45–1.58)	1.93 (0.79–4.68)		Previously not included because of patients less than 100
		OS	1 (1.00–1.00) *	1.09 (0.58–2.05)	2.15 (0.89–5.22)		
Durvalumab/ Sridhar (Sridhar et al., 2019)	Study 1108/ durvalumab monotherapy	Adjusted HR for OS	Ever versus Never: 0.85 by Cox Proportional Hazards Model (Cox Model)			-	-
		HR for	Ever versus Never: 0.85 (OS by Cox Model, Including PD-L1/LM Subgroups)			-	-
		HR for PFS by	Ever versus Never: 0.66 (Cox Model, Including PD-L1/LM Subgroups)			-	-
	ATLANTIC /Durvalumab as third-line or later treatment (53)	Adjusted HR for OS	Ever versus Never: 1.67 (by Cox Model)			-	-
		HR for OS	Ever versus Never: 1.67 (by Cox Model, Including PD-L1/LM Subgroups)				
		HR for PFS	Ever versus Never: 1.08 (by Cox Model, Including PD-L1/LM Subgroups)			-	-

* Assumed values based on information from the study.

## Data Availability

All relevant data are either included in the study or available at databases which are provided in the article.

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
