# Peer review of "From Anti-PD-1/PD-L1 to CTLA-4 and to MUC1—Is the Better Response to Treatment in Smokers of Cancer Patients Drug Specific?"

_jpm, 2021, doi:10.3390/jpm11090914_

Round 1
Reviewer 1 Report
I would like to congratulate the authors to their work and appreciate the chance to review this systematic review. The manuscript provided important insights regarding the efficacy of PD-1 and PD-L1, UTLA-4, MUC1, and EGFR drugs in the treatment of smoker and non-smoker NSCLC and other cancer patients. In my opinion, this manuscript is ready to go. I have only a few suggestions/comments as follows.
- Since this is a systematic review and meta-analysis, a flow chart including the original articles, exclusion criteria, etc. should be helpful for the audience to understand how the eligible and final articles were selected.
- Please kindly check the spelling and English throughout the manuscript. I believe it could be better before it is published.
- I notice some articles have small sample size and may not be representative, should this be announced in the limitation part?
- I feel lost when I was reading the discussion part. Maybe rephrase the structure is a good suggestion for the discussion?
Author Response
Reviewer #1.
I would like to congratulate the authors to their work and appreciate the chance to review this systematic review. The manuscript provided important insights regarding the efficacy of PD-1 and PD-L1, UTLA-4, MUC1, and EGFR drugs in the treatment of smoker and non-smoker NSCLC and other cancer patients. In my opinion, this manuscript is ready to go. I have only a few suggestions/comments as follows.
-Thank you very much for your encouraging comments. We have addressed your suggestions as below.
- Since this is a systematic review and meta-analysis, a flow chart including the original articles, exclusion criteria, etc. should be helpful for the audience to understand how the eligible and final articles were selected.
A1. Thank you for your suggestion. A PRISMA flow chart has been added.
- Please kindly check the spelling and English throughout the manuscript. I believe it could be better before it is published.
Thank you. The English has been checked by Dr. Z. Galvin Li and William Cho.
- I notice some articles have small sample size and may not be representative, should this be announced in the limitation part?
A3. Yes. We have added sentences in the limitation part addressed this issue.
A4. I feel lost when I was reading the discussion part. Maybe rephrase the structure is a good suggestion for the discussion?
We have gone through the discussion again and made minor changes in the sentences.
Reviewer 2 Report
From anti-PD-1/PD-L1 to CTLA-4 and to MUC1, - is the better response to treatment in smokers of cancer patients drug specific?
The finding of this work is merited to be published. The topic is relevant and the results of this study can be useful.
Nevertheless, the main weakness are as follows:
1) Forest plots
- The weights of studies should be added in forest plots.
- The pooled effect size is expressed as mean difference, but in the manuscript authors write hazard ratio. What effect size was calculated? Note: the hazard ratio is not endpoint provided by quantitative analysis, i.e. one of these measures of association: odds ratio, relative risk or difference can be calculated by the meta-analysis.
- The inconsistency (I2) is usually reported with the p-value of the heterogeneity. The authors report the p-value of the pooled result, not that of the heterogeneity. It can lead to misunderstandings.
2) Flowchart of selected studies - is missing.
3) Assessing of selected studies
The methodology of this study does not contain an evaluation of the risk of bias of enrolled publications. It can very significantly affect the final result. Authors should discuss the results with respect to the risk of bias.
4) Tables
Abbreviations are missing
The manuscript requires minor modification.
Author Response
Reviewer #2.
The finding of this work is merited to be published. The topic is relevant and the results of this study can be useful.
Nevertheless, the main weakness are as follows:
-We appreciate very much for your positive comments and addressed your questions and suggestions below.
1) Forest plots
- The weights of studies should be added in forest plots.
- The pooled effect size is expressed as mean difference, but in the manuscript authors write hazard ratio. What effect size was calculated? Note: the hazard ratio is not endpoint provided by quantitative analysis, i.e. one of these measures of association: odds ratio, relative risk or difference can be calculated by the meta-analysis.
- The inconsistency (I2) is usually reported with the p-value of the heterogeneity. The authors report the p-value of the pooled result, not that of the heterogeneity. It can lead to misunderstandings.
A1).
- Per your comments, the weights of studies have been added in forest plots as in the MS. Yes. This information is helpful for the readers.
- You are correct. The pooled effect size was calculated through the difference in means of hazard ratio because the outcome measurements in all studies we observed are made on the same scale.
- Thanks for your point out the I2. We understand in large meta-analyses, I2 can be precise with little bias. Our sample sizes in this study vary a lot. But we also understand that even in small meta-analyses it is better to have a biased and imprecise estimate of I2 than it is to have no estimate at all. In most of the cases, the heterozygosity is low. However, in a few cases, there are high rate of heterozygosity. I assume that you were talking about the Figure 2C. in the random-effects model. I appreciate your carefully observation. Now we have added the potential high heterozygosity exist among these samples. On Page 21, we inserted the following sentences “However, I2 value was 97.35%, suggestion potentially there are heterozygosity among samples. Therefore, further analysis with leave-one-out model was conducted. The leave-one-out model of RE produced the same HR value (Fig. 2D). These data indicate that although heterozygosity may exist among these samples, the difference between smokers and non-smokers is most likely true.”
The second analysis is on Figure 4C. I2 value was 97.36%. On page 25-26, we inserted the following sentence “On one hand, although Random effects model indicated that heterozygosity may influence the data (Figure 4C), leave-one-out model showed the same results (Figure 4D). On the other hand, the sample size was relatively small. We feel confident that future data will confirm our results.”
The third analysis is on Figure 6. On page 33, we added “Because of the potential heterozygosity (Figure 6A) and small sample size, the data cannot be regarded as definitive. However, the chance of the same error in two studies with large number of samples is rare. Future data will most likely confirm this result.”
2) Flowchart of selected studies - is missing.
A2). Thank you. A PRISMA flow chart has been added.
3) Assessing of selected studies
The methodology of this study does not contain an evaluation of the risk of bias of enrolled publications. It can very significantly affect the final result. Authors should discuss the results with respect to the risk of bias.
A3). The protocol for this study has been published. The methodology on the evaluation of the risk of bias of enrolled publications has been mentioned in the protocol. To answer your question, we added the reference of our published protocol. We also added a separate section to further explain this point in the section of methodology.
4) Tables
Abbreviations are missing
The manuscript requires minor modification.
A4). Thanks very much. The abbreviation for overall survival (OS) and Cox Proportional Hazards Model (Cox Model) in Table 1 have been added and used.
Thank you. The English has been checked by Dr. Z. Galvin Li and William Cho.
Additionally, we made change in Graphic Abstract to keep the privacy of the person in the picture.